# Multi-omics analysis in human retina uncovers ultraconserved *cis*-regulatory elements at rare eye disease loci

Victor Lopez Soriano[1,2,6], Alfredo Dueñas Rey[1,2,6], Rajarshi Mukherjee[3], Genomics England Research Consortium*, Frauke Coppieters[1,2,4], Miriam Bauwens[1,2], Andy Willaert[1,2] & Elfride De Baere ®[1,2] ✉

Cross-species genome comparisons have revealed a substantial number of ultraconserved non-coding elements (UCNEs). Several of these elements have proved to be essential tissue- and cell type-specific *cis*-regulators of developmental gene expression. Here, we characterize a set of UCNEs as candidate CREs (cCREs) during retinal development and evaluate the contribution of their genomic variation to rare eye diseases, for which pathogenic non-coding variants are emerging. Integration of bulk and single-cell retinal multi-omics data reveals 594 genes under potential *cis*-regulatory control of UCNEs, of which 45 are implicated in rare eye disease. Mining of candidate *cis*-regulatory UCNEs in WGS data derived from the rare eye disease cohort of Genomics England reveals 178 ultrarare variants within 84 UCNEs associated with 29 disease genes. Overall, we provide a comprehensive annotation of ultraconserved non-coding regions acting as cCREs during retinal development which can be targets of non-coding variation underlying rare eye diseases.

The 'dark matter of the genome' harbors functional *cis*-regulatory elements (CREs) such as promoters, enhancers, silencers, and insulators, whose orchestrated activity is essential to provide spatial and temporal patterns of gene expression that ensure proper tissue development and homeostasis[1,2]. The influence of these regulatory elements on their target genes spans within architectural chromatin units known as topologically associating domains (TADs), demarcated by boundaries enriched in CTCF and cohesin proteins[3]. Due to their high context specificity, the annotation of candidate CREs (cCREs) poses an attractive yet challenging task. In this regard, the Encyclopedia of DNA Elements (ENCODE)[4,5] represents a powerful tool to characterize functional cCREs, as it provides a robust inventory of well-defined cCREs supported by epigenetic data derived from a wide variety of both human tissues and cell types.

Over the past decades, comparative genomics have been made progress to map functional cCREs by establishing cross-comparison between species. As a result, several databases of non-coding sequences that exhibit extremely high conservation have been generated, including UCEs[6,7], UCNEs[8], ANCORA[9], and CONDOR[10]. A direct comparison of these resources is, however, not straightforward due to the intrinsic differences in the scopes of these databases. One of the most comprehensive resources is UCNEbase, which comprises exclusively >200-bp-long non-coding genomic regions that exhibit ≥95% sequence identity between human and chicken. The selection of these two species was based on both biological and technical arguments, namely their considerable evolutionary distance enhancing the accuracy in identifying functional elements, and the high-quality of their respective genome assemblies. In total, 4351 regions across the genome comply with the criteria and are listed as ultraconserved non-coding elements (UCNEs)[8]. The vast majority of these highly constrained elements cluster around key developmental genes, such as *PAX6*, and have been long hypothesized to act as tissue-specific

[1]Department of Biomolecular Medicine, Ghent University, Ghent, Belgium. [2]Center for Medical Genetics, Ghent University Hospital, Ghent, Belgium. [3]Department of Ophthalmology, St James's University Hospital, Leeds, UK. [4]Department of Pharmaceutics, Ghent University, Ghent, Belgium. [6]These authors contributed equally: Victor Lopez Soriano, Alfredo Dueñas Rey. *A list of authors and their affiliations appears at the end of the paper. ✉ e-mail: elfride.debaere@ugent.be

transcriptional regulators[8,11]. Ultraconserved CREs have been shown to be consistently depleted of common variants[12,13], indicating that purifying selection has shifted ultraconserved CRE-derived allele frequencies towards magnitudes similar to those observed for missense variants[13], hence reinforcing that variation within these regions is more likely to have functional consequences.

Due to the wide implementation of whole genome sequencing (WGS) in human genetic studies and ambitious initiatives such as the 100,000 Genome Project (100kGP) launched by Genomics England (GEL)[14], previously overlooked regions within the vast non-coding fraction of the genome have been investigated in patients with rare diseases and have shown an emerging role of non-coding variants in disease pathogenesis. Two remarkable examples of functional evidence of disease driven by disruption of conserved CREs can be found precisely in the rare eye disease field, in which mutations located within highly conserved non-coding regions have been linked to developmental defects. A first example is a de novo single nucleotide variant (SNV) in an ultraconserved enhancer (SIMO element) 150 kb downstream of an intact *PAX6* transcriptional unit, found in a case with aniridia[15]. Second, tandem duplications within a gene desert downstream the *IRXA* cluster[16,17] have been found in patients affected by North Carolina Macular Dystrophy (NCMD); the shared duplicated region harbors an enhancer element defined as UCNE that could act as a *cis*-regulator during retinal development[18].

The retina is a heterogeneous tissue composed of neuronal (rod and cone photoreceptors, bipolar cells, ganglion cells, horizontal cells, and amacrine cells) and non-neuronal cell types (astrocytes, Müller cells, and resident microglia) that arise from the common pool of retinal progenitor cells (RPCs) (with the exception of astrocytes and microglia)[19,20]. This cellular complexity is the result of spatiotemporally controlled gene expression programs during retinal development, requiring concerted action of thousands of CREs[21,22]. In the wide spectrum of retinopathies, inherited retinal diseases (IRDs) represent a leading cause of early-onset vision impairment affecting over 2 million people worldwide. Despite targeted panel-based sequencing and whole exome sequencing (WES)-based genetic testing 30–50% of IRD cases remain unsolved[23,24], raising the hypothesis that disease-causing mutations are found in the non-coding genome that are mostly not covered by standard genetic testing.

In this study, we set out to annotate UCNEs as cCREs that modulate gene expression during retinal development (Fig. 1a) and that can play a role in retinopathies such as IRDs. Firstly, we performed a comprehensive integration of publicly available multi-omics datasets derived from human retina based on biochemical features associated with regulation of gene expression. Secondly, UCNEs co-localizing with genomic regions showing regulatory activity were linked to target genes that exhibit expression in the retina. Finally, we evaluated the contribution of genomic variation within these regions to rare eye diseases. This allowed us to identify an ultrarare SNV in a candidate *cis*-regulatory UCNE located upstream *PAX6* in a family displaying autosomal dominant foveal abnormalities, for which we provide functional evidence based on transgenic enhancer assays in zebrafish. Overall, this work has improved the functional annotation of UCNEs in human retina representing understudied targets of non-coding variation that may explain missing heritability in rare eye diseases.

## Results
### Integration of bulk and single-cell multi-omics data enables the identification of 1487 UCNEs with a candidate *cis*-regulatory role in human retina

The wealth of publicly available data allowed the integration of transcriptomic (bulk and scRNA-seq) and epigenomic (DNase-seq, scATAC-seq, ChIP-seq) data to evaluate the regulatory capacity of UCNEs over the major developmental stages of human retina. More specifically, to identify which UCNEs might have a potential *cis*-regulatory role, an intersection was performed with genomic regions characterized by an open chromatin context in the retina at different developmental stages, as supported either by DNase (ENCODE) or scATAC-seq[25] peaks. This resulted in a total of 1487 UCNEs, i.e., approximately one-third of UCNEs display a retinal open context (Fig. 1b, Supplementary Data 1) (hereinafter referred to as putatively active UCNEs). Approximately 80% of the DNase-seq peaks were also supported by scATAC-seq, hence defining high-confidence candidate *cis*-regulatory UCNEs (Fig. 1b). Nevertheless, to account for differences in time points between datasets, all 1487 UCNEs were included in downstream analyses. Apart from the display of a retinal open chromatin context, we also evaluated the overlap between UCNEs and genomic regions featured by markers associated with regulation of gene expression. A total of 834 UCNEs were found to display at least one of the assessed features in the interrogated retinal developmental stages; more specifically 111 UCNEs were identified to display the active enhancer mark H3K27ac, of which most (95%) also exhibit other histone marks related to active enhancers (H3K4me1 and H3K4me2) (Fig. 1b). Out of these 111 UCNEs, 33 were found to maintain signatures consistent with enhancer activity (H3K27Ac) at adult stage. Table 1 and Supplementary Data 1 and 2 include summarized and detailed overviews of the number of identified UCNEs per marker and stage, respectively.

To investigate the functional relevance of the putative regulatory UCNEs, we performed a correlation with the 1942 experimentally validated human non-coding fragments of the VISTA Enhancer Browser (Fig. 1c). Out of these 1942 elements, approximately 50% (998/1942) display enhancer activity. When overlapped with all 4135 UCNEs, a slight increase of positive elements (505/922) is observed; however, when only the identified retinal accessible UCNEs were considered, roughly 68% were found to be positive (272/402), thereby reinforcing their predicted regulatory role (Fig. 1c). Interestingly, evaluation of the anatomical description of the reporter gene expression patterns of these 272 elements revealed a potential enrichment in eye (Fig. 1d). An illustrative example of one of these elements and its corresponding multi-omics-based characterization is shown in Fig. 1e, f.

### In silico prediction and integration of retinal chromatin conformation data identify putative target genes under UCNE *cis*-regulation

We made use of the *GREAT* algorithm to assign potential target genes to the identified putatively active UCNEs and thus assess their association with genes expressed in the retina[26]. A total of 724 target genes were assigned to the initially identified 1487 UCNEs displaying candidate *cis*-regulatory activity. The vast majority (74.5%) of queried regions were assigned as putatively regulating two genes based on the used association rule (Fig. 2a). Furthermore, most of these regions were found to be located at 50-500 kb from the TSS of the regulated target genes regardless of their orientation (Fig. 2b). Out of these 724 target genes, 594 (81.9%) are expressed in the retina in at least one of the interrogated developmental stages (52 to 136 days post-conception). To evaluate further whether chromatin contacts between the UCNEs and the promoters of their target genes can occur in retina, we integrated the 2948 retinal TADs identified by Marchal et al.[27]. A large majority (393/594) of the target genes were found to be in TADs also harboring their associated UCNE (Supplementary Data 1). These genes were then used as input for Gene Ontology analysis, which revealed an overrepresentation of terms related to regulation of transcription and differentiation, thereby confirming the expected roles of UCNEs as tissue-specific CREs during development, particularly of the nervous system (Fig. 2c; Supplementary Data 3).

As a last layer of characterization of these putatively active UCNEs and their candidate target genes, we assigned to each target gene cell-type specificity information based on expression data. Out of the 1487 putatively active UCNEs, 808 (54.3%) displayed a cell-type-specific open chromatin context consistent with their target gene(s)

**a** Annotation of ultraconserved non-coding regions acting as candidate *cis*-regulatory elements

Retrieval of ultraconserved non-coding elements

Retinal bulk & single-cell multi-omics dataset integration

Prediction of UCNEs with potential *cis*-regulatory role

**b**

**c**

**d**

**e** VISTA [hs1170]

**f**

expression signature(s) (Supplementary Data 1). Furthermore, we evaluated the overlap between the sets of target genes assigned to UCNEs by *GREAT* followed by gene expression filtering and those mapped by implementing the peak-to-gene linkage method on the scATAC-seq data. Although a substantial number of target genes overlapped between both methods (Supplementary Data 4), it is noteworthy to mention that almost half of the UCNEs were assigned to different target genes, most likely due to differences in the annotations employed by both methods.

## Mining of WGS data from a rare eye disease cohort reveals rare variants within putatively active UCNEs

Out of the 594 putative target genes under UCNE *cis*-regulation, 45 were found to be previously linked to a rare eye disease phenotype, of

**Fig. 1 | Integration of publicly available datasets for the characterization of the ultraconserved non-coding elements (UCNEs) library. a** Overview of the integrative multi-omics analysis for the prediction of UCNEs with potential *cis*-regulatory role in human retina. **b** Venn diagrams illustrating the CREs displaying open chromatin features based on scATAC-seq and DNase-seq (left) and their overlap of active enhancer marks (H3K27ac, H3K4me1, and H3K4me2) (right). **c** Barplot showing the proportions of elements from the VISTA enhancer browser (V), UCNEs (U), and putative regulatory UCNEs (characterized by retinal datasets) (U⁺) displaying reporter expression (positive). **d** Proportional stacked barplot showing the distribution of tissues (eye, forebrain, hindbrain, midbrain, neural tube, limb, and tail) in which the putative regulatory UCNEs display reporter expression. **e, f** Illustration of one of the characterized UCNEs (*NR2F1_Hector_1/2*) displaying open chromatin supported by DNase-seq (embryonic day 74–85, 89, and 103–125) and scATAC-seq (AC/HC/GC Precursors Cells, Early Progenitor Cells, Ganglion Precursor Cells, Late Retinal Progenitor Cells) and enhancer reporter expression in the eye. Figures obtained from VISTA (hs1170) enhancer (E) and UCSC genome (F) browsers. FW: fetal weeks; V: VISTA enhancer elements; U: UCNEs; U⁺: Putative active regulatory UCNE; ∩: intersection. Figure 1a was created with BioRender.com.

which 23 are IRD genes and 19 are also associated with other developmental disorder phenotypes (Supplementary Data 5). Considering the extreme selective constraints of UCNEs, we hypothesized that genetic changes in these regions could contribute to disease. Therefore, we evaluated the genomic variation within the putatively active UCNEs associated with these 45 disease genes in a sub-cohort of individuals affected by rare eye diseases ($n = 3206$, Supplementary Data 6) from the 100,000 Genomes Project (Fig. 3a), of which 1802 cases were unsolved.

As expected, a depletion of common variation (MAF > 1%) was observed across these regions (Fig. 3b). This depletion was found to be not only restricted to the examined sub-cohort but rather constitutes a more general phenomenon. In particular, UCNEs were found to exhibit significantly lower genome-wide residual variation intolerance score (gwRVIS)[28] values compared to a set of randomly selected genomic regions, thereby reflecting their high intolerance to genomic variation (Fig. 3c).

A total of 431 (426 SNVs and 5 SVs) ultrarare variants, i.e., absent from reference population public databases, were identified in 199 putatively active UCNEs linked to these 45 genes (Supplementary Data 7). Of these variants, 109 were found in unsolved cases. In addition, we computed the allele frequency distribution of all variants retrieved within a selection of 25 of these disease genes and compared it against that of their corresponding UCNE. As before, a distinct depletion of common variation was observed, thereby further supporting the specificity of the overlap of (ultra-)rare variants and UCNEs (Supplementary Figure 1).

Notably, out of the 5 identified SVs, one corresponds to the known shared duplicated region downstream of *IRX1*, located within the NCMD-linked *MCDR3* locus [MIM: 608850]; in particular, this duplication, identified in 8 affected individuals of 4 different families segregating macular defects consistent with NCMD, involves a UCNE (*IRXA_Aladdin*) exhibiting chromatin accessibility in developing horizontal cells. Finally, out of these ultrarare variants, 178 are located within 84 UCNEs displaying histone modification marks (in at least one

of the interrogated stages), associated with 29 genes. This set defined our primary search space for variants with potential functional effects and further assessment.

## An ultrarare SNV in an active UCNE upstream of *PAX6* found in a family segregating autosomal dominant foveal anomalies

We identified an ultrarare SNV (chr11:31968001 T > C) within a candidate *cis*-regulatory UCNE located -150 kb upstream of *PAX6* (*PAX6_Veronica*). This variant was found in four individuals of a family segregating autosomal dominant foveal abnormalities (Supplementary Figure 2A, B; Supplementary Data 8). A *CFH* missense variant (c.1187 A > G, p.Asn396Ser) was initially reported in the affected individuals but could not explain the foveal anomaly. Given the phenotype, a variant screening was performed with a particular focus on the *PRDM13* and *IRX1* NCMD loci, both associated with foveal or macular mis-development. A total of three and two SNVs within the *IRX1* and *PRDM13* loci, respectively, were found to segregate with disease. However, these SNVs are present in individuals from reference population databases and are not located in genomic regions displaying *cis*-regulatory features (Supplementary Data 9). Equally, loci associated with foveal hypoplasia, nystagmus and hypopigmentation (*AHR*, *FRMD7*, *GPR143*, and *SLC38A8*) were assessed for SNV or SV segregating with disease. No other (likely) pathogenic variants in these loci or any other known IRD/rare eye disease gene were identified in the affected cases (Supplementary Data 9).

The identified chr11:31968001 T > C variant affects a nucleotide residue that is conserved for at least 360 million years of evolution that separate humans from *X. tropicalis*. In silico assessment of this variant and flanking sequence pointed to a likely deleterious effect and revealed a predicted disruption of several TF binding motifs (Supplementary Data 9). More specifically, this variant is located within a UCNE (*PAX6_Veronica*) that is cataloged as a cCRE in ENCODE (EH38E1530321), featured by distal enhancer-like signatures in bipolar neurons. Regarding its retinal context, this UCNE displays accessible chromatin in the early stages of retinal development (7/8 gestational weeks), and in particular in ganglion cell precursors (Fig. 4a, b). Importantly *PAX6* is expressed within this pool of cells and, interestingly, this expression appears to be enriched within a specific subpopulation, as observed from the distribution of expression values across all the cells composing this cluster (Supplementary Figure 3). In addition, this UCNE was found to display the active enhancer mark H3K4Me1 at the earliest time points available for this dataset (13/14 and 15/16 gestational weeks).

This region has also been functionally validated in transgenic mouse assays (VISTA element hs855), which revealed gene enhancer activity mainly in the forebrain and, to some extent, in the retina (3/6 embryos) (Fig. 5a). A closer assessment of this UCNE with regard to its proximity to the *PAX6* promoter in other species revealed its syntenic conservation up to zebrafish; interestingly, this element localized in closer proximity to the *PAX6* promoter in species like *X. tropicalis* (60 kb distance from the TSS) or *Danio rerio* (15 kb distance from the *pax6a* TSS) (Supplementary Figure 4).

**Table 1 | Overview of the number of events based on the peak identification for each marker at the different stages of retinal differentiation**

| ChIP-seq | Stage | | | | |
|---|---|---|---|---|---|
| | FW13/14 | FW15/16 | FW18/20 | FW23/24 | Total Unique |
| H3K27ac | 73 | 45 | 74 | 94 | 127 |
| H3K4me1 | 235 | 490 | 48 | 323 | 582 |
| H3K4me2 | 347 | 354 | 256 | 225 | 465 |
| H3K4me3 | 94 | 124 | 102 | 97 | 136 |
| H3K27me3 | 119 | 54 | 128 | 126 | 154 |
| H3K9/14Ac | 51 | 76 | 36 | 31 | 94 |
| H3K9me3 | 0 | 1 | 1 | 29 | 29 |
| Pol II | 17 | 66 | 59 | 55 | 106 |
| CTCF | 51 | 69 | 22 | 37 | 71 |

*FW* fetal weeks.

**a**

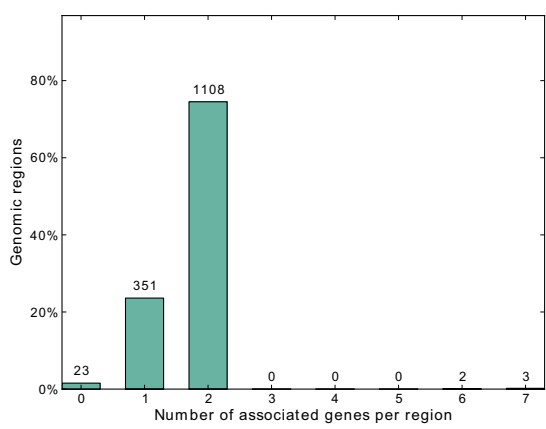

**b**

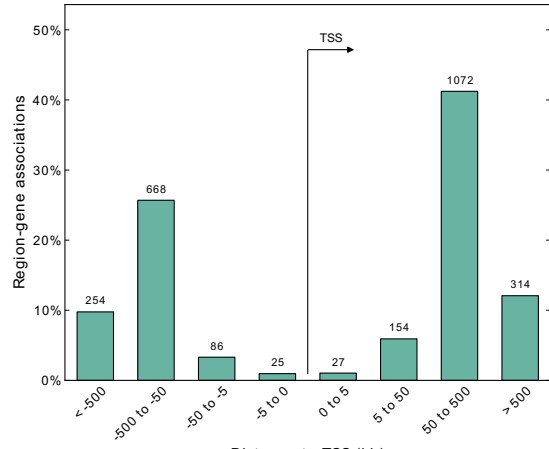

**c**

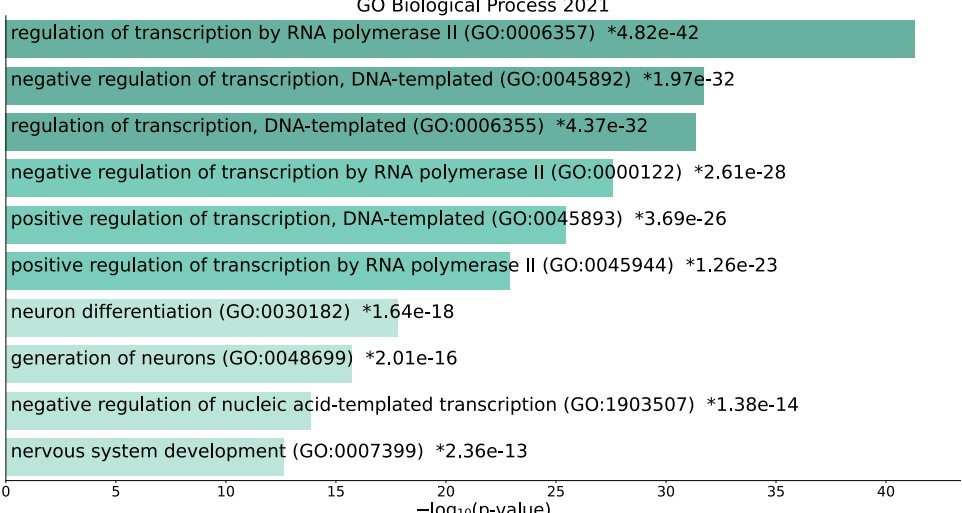

**Fig. 2 | Target genes under putative regulation of characterized UCNEs in retina. a** Barplot showing the number of associated genes per retinal UCNE. **b** Distance distribution from the TSS and its associated UCNE. **c** Gene ontology for the UCNE-associated target genes. An adapted Fisher's exact test as in Chen et al. (*EnrichR*)[98] was performed, see also Supplementary Data 3. GO Gene Ontology, TSS transcription start site.

## In vivo enhancer assays in zebrafish support the association between candidate *cis*-regulatory UCNE and *PAX6*

Given the putative retinal activity suggested by the VISTA enhancer assay for this UCNE, we performed zebrafish enhancer assays to gain further insights into its regulatory range of action. We found consistent reporter expression in several regions, including the forebrain, hypothalamus/otic vesicle, somites, nosepit, and eye. Importantly, we observed that the GFP expression in the eye initiated in a small fraction of the embryos at 2 dpf (4/22 GFP⁺ embryos) increasing to a larger proportion of the embryos by 3 dpf (21/22 GFP⁺ embryos) and eventually disappeared from 4 dpf (Fig. 5b). Altogether, these observations are consistent with the epigenomic characterization based on retinal datasets and provides further evidence of a potential *cis*-regulatory role of this UCNE on *PAX6* expression. Using the same setting, we also evaluated the reported expression activity the mutated version of this UCNE, i.e., harboring the SNV identified in the affected individuals of the family described before. However, no conclusive differences were observed with respect to the wild-type version. A detailed overview of the results of these assays can be found in Supplementary Data 10.

## Discussion

Since the landmark study of Bejerano et al.[7] almost twenty years ago, ultraconservation of non-coding elements and their paradoxical functional (in)dispensability[13,29–32] have equally stirred controversy and fascination among scientists from diverse disciplines. A recent study demonstrated that UCNEs with robust enhancer activity during embryonic development appear to be unexpectedly resilient to mutation[29]. There are, however, clear instances in which variation within ultraconserved CREs can be a driver of human rare disease[15,33–40]. Despite the substantial body of literature on these intriguing elements, thus far there are no comprehensive studies based on the integration of multi-omics to map the regulatory capacity of UCNEs in a specific tissue or cells, and on human genetic data to assess their contribution to disease. Apart from a few well-known examples, variants in CREs are an underrepresented cause of Mendelian diseases. A major obstacle hampering their identification is the need to define the search space of the CREs they affect in a tissue- and even cell-type-specific manner. Here we set out to address this challenge in human retina, of which the *cis*-regulatory architecture has been well-studied[25,27,41] and for which there is emerging *cis*-regulatory variation implicated in disease[15,18,33,37].

**a**

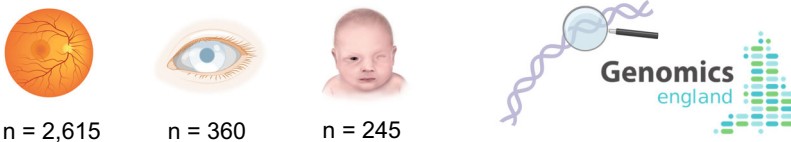

**Mining of variants within candidate *cis*-regulatory UCNEs in WGS data of a rare eye disease cohort**

n = 2,615        n = 360        n = 245

**b**

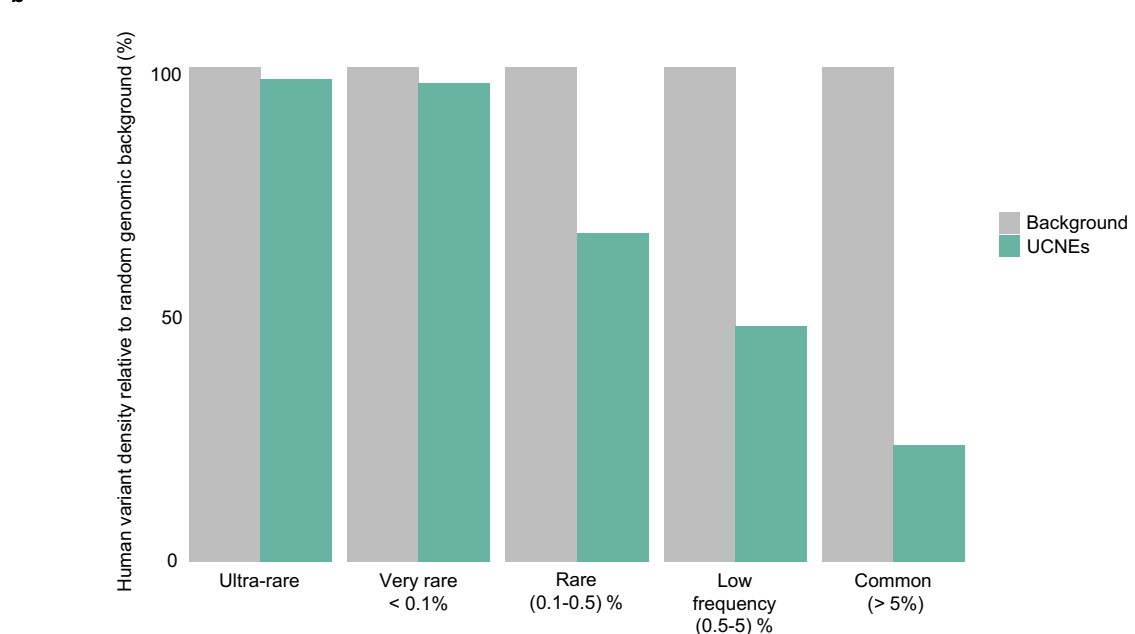

**c**

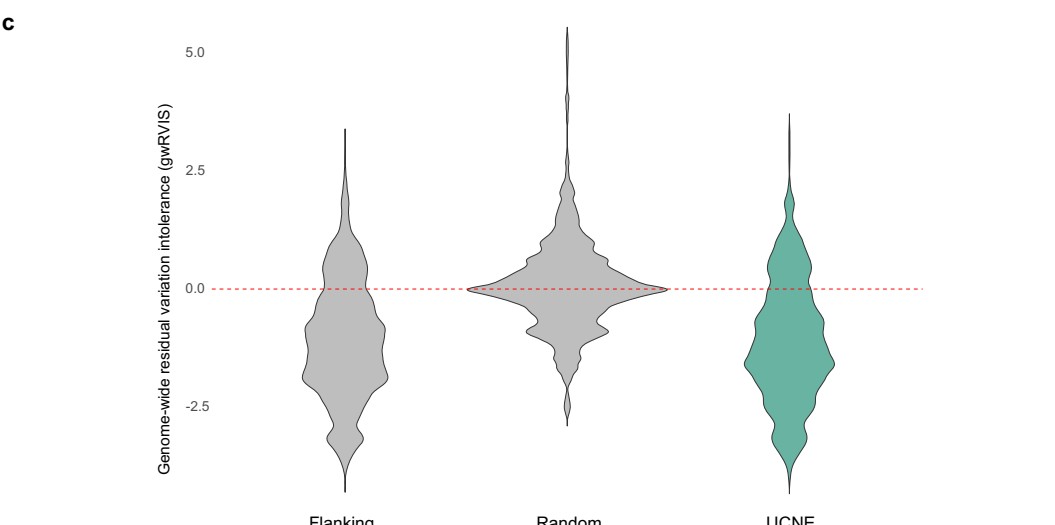

**Fig. 3 | Contribution of UCNE genomic variation to missing heritability in rare eye diseases. a** Overview of a sub-cohort comprising *n* = 3206 participants of the 100,000 Genomes Project affected by posterior segment abnormalities (*n* = 2615), anterior segment abnormalities (*n* = 360), and ocular malformations (*n* = 245). **b** Variant population frequencies within putative retinal UCNEs normalized to a background composed of randomly selected sequences (see "Methods").

**c** Significantly lower genome-wide residual variation score (gwRVIS) values scores are observed across UCNEs and their flanking regions compared to the background composed randomly selected sequences, thereby showing the high intolerance to variation of UCNEs (*$p < 0.001$, Kruskal–Wallis rank sum test followed by post-hoc Dunn test with Bonferroni correction; Z-scores = −169.33 (Flanking-Random), −170.36 (UCNE-Random). Figure 3a was created with BioRender.com.

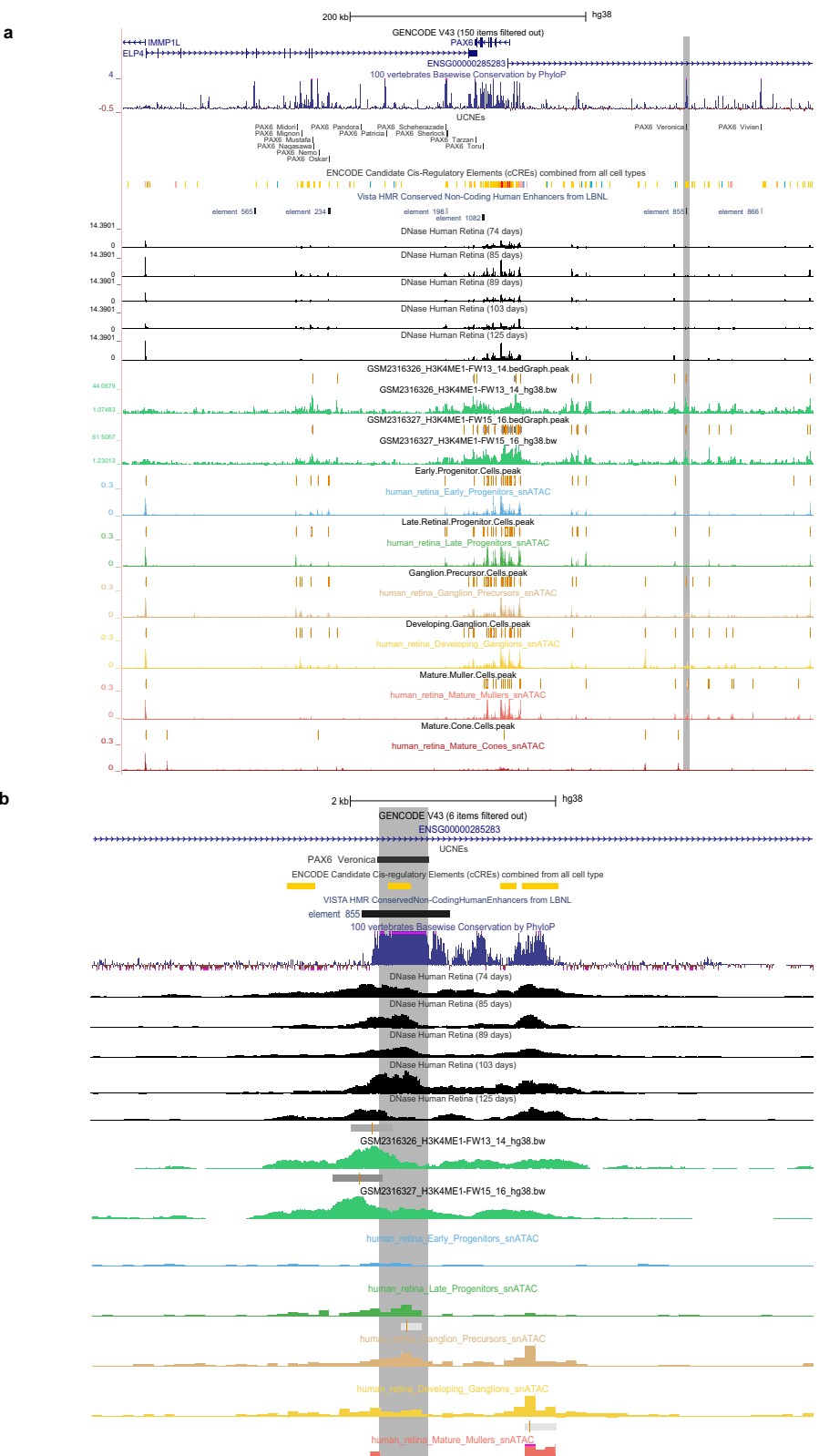

**Fig. 4 | Characterization of the *cis*-regulatory landscape of *PAX6* in human retina. a** Visualization of the epigenomic context of the identified *PAX6*-associated UCNE (highlighted in gray) in relation to the *PAX6* locus (top; chr11:31,490,261–32,075,591) and **b** Zoomed-in of the identified element (bottom; chr11:31,965,000–31,972,000). Figures are obtained from the UCSC genome browser.

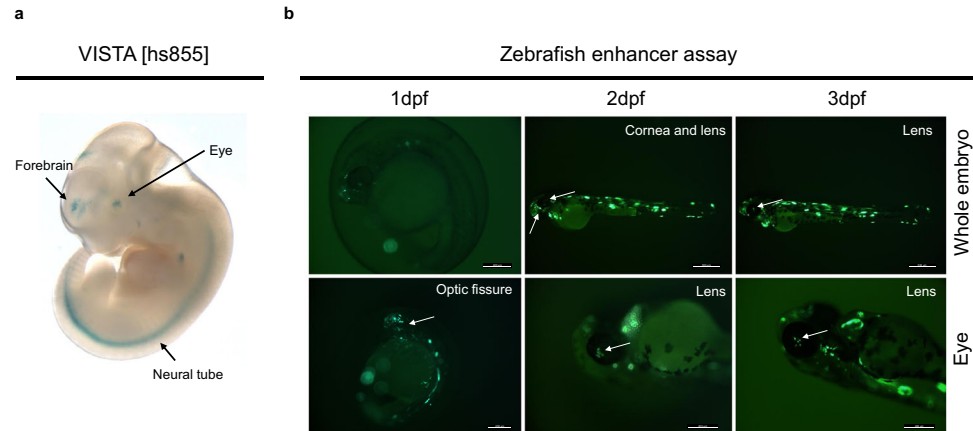

**Fig. 5 | In vivo evaluation of the _PAX6_-associated UCNE. a** This UCNE displays reporter activity in the human embryonic forebrain (6/9 embryos) as well as in other structures including eye (3/6 embryos) and neural tube. Figure obtained from the VISTA enhancer browser (hs855). **b** Zebrafish enhancer assays (_n_ = 2, see also Supplementary Data 10) at 3 different time points (1 dpf, 2 dpf, 3 dpf) showing GFP-positive tissues (optic fissure −1 dpf-, lens cells −2,3 dpf-, and forebrain −2,3 dpf-; white arrows). dpf days post-fertilization.

Although ENCODE provides a well-annotated inventory of cCREs, its retinal datasets do not include the earliest stages of RPC differentiation. To overcome this and to incorporate cell-type specificity information, we integrated scATAC-seq, which allowed us to identify 1487 UCNEs characterized by accessible chromatin in at least one major stage of retinal development. We also interrogated the epigenetic landscape of these elements by analyzing histone modification patterns associated with e.g., active/poised enhancers (H3K27ac, H3K4me1, H3K4me2)[42–44] and highly packed chromatin (H3K27me2 and H3K27me3)[45]. This analysis provided a useful approach to trace the activity of elements that are active at a specific stage during retinal development. In total, 111 UCNEs were found to display active enhancer marks during retinal development. Similarly to ENCODE, the analyzed ChIP-seq datasets do not cover the earliest stages and hence these 111 elements are more likely to be related to stages of differentiation corresponding to later-born cell types (i.e., rod photoreceptors, bipolar cells, Müller glial, and some varieties of amacrine cells)[46]. Interestingly, of these 111 elements, 33 were found to exhibit sustained enhancer-related signatures in adult stage, suggesting their potential role in the maintenance of gene expression patterns after development. Overall, our data integration spans all the major stages of retinogenesis and thus provides a comprehensive framework for the systematic characterization of ultraconserved retinal CREs.

Establishing an association between cCREs and their putative target genes is essential for the interpretation of genomic variation that could disrupt cell type-specific binding sites of TFs and/or long-range chromatin contacts that can lead to ectopic expression of genes[47,48]. Here we combined _GREAT_[49] with retinal expression and Hi-C-derived retinal data to assign a range of action to candidate _cis_-regulatory UCNEs. We found this procedure to be more comprehensive than assigning target genes only based on the correlation provided by the peak-to-gene linkage method we used[26]. In total, 594 retina-expressed genes can be under UCNE _cis_-regulation; interestingly most of the UCNEs were found to be distributed in distal position from the TSS of their assigned target genes, as observed for enhancers promoting expression of developmental genes[39,50,51]. In this regard, a large fraction of the identified ultraconserved putative enhancers clusters around key developmental genes (e.g., _MAB21L2_, _OTX2_, _PAX6_, _SOX2_, _ZEB2_) known to be controlled by complex regulatory landscapes[11]. Besides, 45 of these genes are implicated in rare eye diseases. Interestingly, coding variants in these genes lead in most cases to syndromic developmental phenotypes with other manifestations apart from the ocular ones as collated in the G2P database[23]. As it has been

shown before, the phenotype caused by a coding mutation of a developmental gene can be different from the phenotype caused by a mutation in a CRE controlling spatiotemporal expression of this gene; this is exemplified by the _PRDM13_ locus, for which bi-allelic coding _PRDM13_ variants result in hypogonadotropic hypogonadism and perinatal brainstem dysfunction in combination with cerebellar hypoplasia[52] while _cis_-regulatory variants in retinal CREs leads to NCMD, a developmental macular disease[18]. From our analyses we could identify a cCRE for _PRDM13_ (_PRDM13_Blooklyn_) that displays an open chromatin context in early RPCs and developing amacrine cells, which are precisely the retinal cell types in which _PRDM13_ is expressed[53,54]. Although no disease associations have been established thus far for other genes for which we identified candidate _cis_-regulatory UCNEs, some of them are known to play important roles in retina development, e.g., _NFIB_[55], and therefore represent potential target regions for non-coding variants contributing to missing heritability.

A limitation of this analysis is the usage of a pre-defined set of genomic regions, in particular UCNEs, as this database is static and rarely updated[11]. Our approach could thus be expanded using more flexible criteria, thus including other less conserved, albeit disease-relevant loci[33]. Nonetheless, the advantage of working with highly conserved genomic regions is the availability of already validated experimental data[56]; additionally, as shown here, the integration of tissue-specific epigenomic data provides a robust framework to identify potential bona fide enhancers and their corresponding target genes, and clinically relevant variants therein.

To further investigate the contribution of genomic variation within candidate _cis_-regulatory UCNEs to disease, we mined WGS data from a sub-cohort of 100,000 Genome Project Genomics England participants affected by different rare eye diseases. As reported previously[29,57], we observed a substantial depletion of common variation across these regions when compared to a background comprising randomly selected genomic regions. Indeed, intolerance to variation was also found to extend to the regions flanking the ones strictly defined as UCNEs. This apparently strong purifying selection has been proven difficult to reconcile with recent findings demonstrating remarkable resilience of UCNEs to variation[12]. Moreover, it has been suggested that weaker but uniform levels of purifying selection across hundreds of bases and different species could bring together these otherwise contradictory observations and explain why rare variants are not significantly depleted within UCNEs[58]. Our primary search space for variants with potential functional effects

comprised 178 ultrarare variants located within 84 putatively active UCNEs associated with 29 genes. Interestingly, most of them were predicted to have a likely functional and/or deleterious effect, which could be explained by a skewed cumulative importance towards evolutionary conservation-related features in the predictive models[59–62].

Out of all variants found, one ultrarare variant (chr11:31968001 T > C) within a UCNE displaying open chromatin in progenitor ganglion cells and linked to *PAX6* was further dissected. This variant was exclusively found within the studied family, displaying isolated foveal abnormalities, and initially solved with an ultrarare missense variant in *CFH*, which could not fully explain the clinical presentation[63,64]. As no other likely pathogenic variants were found within relevant loci[17,18,65–67], and the observed phenotypes of the affected individuals match within the *PAX6* disease spectrum, mis-regulation of *PAX6* expression cannot be excluded as a pathogenetic mechanism. Genetic defects of *PAX6* have been found in aniridia, a pan-ocular disorder characterized by the absence or hypoplasia of the iris, nystagmus and foveal hypoplasia[68], the latter comprising thinning of macular inner and outer retinal layers consistent with misdirected foveal development[69]. Recently, a phenotype characterized by isolated foveal hypoplasia with nystagmus has been also linked to *PAX6* variation[70]. Given the known inter- and intra-familial phenotypic variability observed in *PAX6*-associated disorders[71,72], variable expressivity cannot be discarded for the identified variant. Thus far, in terms of regulatory variation in CREs, only a single SNV located within the ultraconserved SIMO element has been associated with *PAX6*-disease. This subtle change (SNV), identified de novo in an individual with aniridia and foveal hypoplasia, was found to disrupt an autoregulatory PAX6 binding site[15]. Importantly, we could establish in vivo, using the zebrafish as animal model, a developmental expression pattern in the eye driven by this UCNE in tissues for which *PAX6* expression is relevant[73–77] and within a time window consistent with the period in which the zebrafish retina has become fully laminated[78,79]. We could not obtain conclusive functional results for the identified variant using our experimental approach. A major limitation of this setting is the fact that the UCNE and its mutant version were tested outside of their native genomic context. Therefore, to validate the role of this UCNE in *PAX6* regulation, further functional assays that consider the native context, such as CRISPRi[80], are needed. Studying the molecular consequences of the variant itself in a patient-derived cellular model, however, is more challenging, since currently no models, including patient-derived retinal organoids, fully recapitulate foveal patterning[81].

Overall, our work is exemplar for how the wealth of publicly available multi-omics data can be used to exploit the regulatory capacity of UCNEs in a tissue- and cell type-specific way. As demonstrated here, UCNEs can represent understudied regions of noncoding variation underlying missing heritability in Mendelian diseases. With the increasing implementation of WGS in rare disease research and diagnosis, the delineation of tissue and cell-type-specific CREs will be a prerequisite to identify and fully interpret the pathogenic nature of non-coding variants. With this study, we have illustrated how the creation of a comprehensive set of functionally annotated UCNEs in a target tissue can represent a powerful initial strategy to narrow down the variant search space, particularly for developmental phenotypes.

## Methods

### Ethics approval and consent to participate

The 100,000 Genomes Project Protocol has ethical approval from the HRA Committee East of England – Cambridge South (REC Ref 14/EE/1112). This study was registered with Genomics England under Research Registry Projects 465.

## Integration of UCNEs with bulk and single-cell epigenomic, regulatory and transcriptional datasets from human developing and adult retina

The 4351 genomic regions defined as UCNEs[8] were used as the basis for the integration of multiple publicly available multi-omics datasets derived from embryonic and adult human retina. More specifically, to evaluate the potential function of UCNEs as *cis*-regulatory modules, we made use of DNase-seq[5], ChIP-seq of histone modifications[41,82], and single-cell (sc) ATAC-seq[25] derived from retinal tissue collected at distinct stages of development. Furthermore, to correlate potential *cis*-regulatory activity with gene expression, we mined both bulk[83] and scRNA-seq[25] generated at stages overlapping or extending the ones of the epigenomic datasets. In addition, as a last layer of functional characterization, we integrated into our analyses the experimental data of the VISTA Enhancer Browser[56], which allows the classification of candidate regulatory elements based on in vivo enhancer reporter assays tested in transgenic mice at embryonic day 11.5[84]. Supplementary Data 11 provides an overview of all used datasets.

## Identification of candidate *cis*-regulatory UCNEs in retina

Data generated by scATAC-seq of embryonic (53, 59, 74, 78, 113, and 132 days) and adult (25, 50, and 54 years old) human retinal cells were obtained (GSE183684) and imported into R (v4.0.5). Count matrices were processed using the *ArchR* single-cell analysis package (v1.0.1)[26] as described in Thomas et al.[25], with minor modifications. Briefly, the total number of cells after filtering out doublets was 61,313. Single-nucleus RNA-seq of the same tissue and time points (GSE183684) were integrated using the unconstrained integration method so as to establish a linkage between the scATAC-seq and scRNA-seq datasets. More specifically, the unconstrained integration is a completely agnostic approach considering all cells derived from an scATAC-seq experiment and attempting to align them to any cells in the respective scRNA-seq experiment[26].

Both datasets were then used to assign retinal cell class identities to the different clusters based on known markers[25] and subsequent peak calling. BigWig files from each annotated cell cluster were extracted and converted into bedGraph files using *bigWigtoBedGraph* UCSC utility; narrow peak detection was performed using *bdgpeakcall* (MACS2.2.7.1)[85] with default parameters and a value of 0.4 as cut-off (median peak width: 300 bp). Similarly, bigWig files corresponding to histone modification patterns (H3K27Ac, H3K27me3, H3K36me3, H3K4me1, H3K4me2, H3K4me3, H3K9/14Ac, H3K9me3, PolII, CTCF) were retrieved (GSE87064) and converted into bedGraph files; in this case, the cut-off value for peak calling by *bdgpeakcall* was set at 20. In addition, the subset of ENCODE DNase hypersensitivity sites (rDHSs) identified in embryonic retina (74–85, 89, 103–125 days) were obtained (ENCSR786VSQ, ENCSR666FML, ENCSR632UXV; median peak width: 300 bp) and elements featured by *Low-DNase* filtered out.

Finally, to identify putative *cis*-regulatory UCNEs, we screened for overlaps between UCNEs and the retrieved sc-ATAC/ChIP/DNase-seq peaks using *bedtools intersect* (*BEDTools* v2.30.0)[86] with default parameters; of note *bedtools window* including a ±250-bp-long window was used when overlapping the ChIP-seq peaks to characterize more broadly the chromatin status in the vicinity of the accessible UCNEs (Supplementary Figure 5).

## Identification and characterization of target genes under putatively active UCNE regulation

To assign potential target genes to the identified active UCNEs we used the Genomic Regions Enrichment of Annotations Tool (*GREAT*)[49] (v4.0.4). Briefly, this tool computes statistics by associating genomic regions with nearby genes and applying the gene annotations to the regions. More specifically, when run against a whole genome background, two statistical tests are performed, namely the binomial test over genomic regions and the hypergeometric test over genes, thereby

providing comprehensive annotation enrichments for the input genomic regions. Here, *GREAT* was run using an association rule based on the definition of an extended basal regulatory domain. Each gene in the genome was assigned a basal regulatory domain (5 kb and 1 kb upstream and downstream of the transcription start site {TSS}, respectively) with an extension of up to 1 Mb to the nearest gene's basal domain. Each putatively active UCNE was then associated with all genes whose regulatory region it overlapped. In addition, curated regulatory domains were also included. These domains are supported by experimental evidence demonstrating that a gene is directly regulated by an element located beyond of its putative regulatory domain. In particular, for the utilized version of *GREAT*, these domains included the Sonic Hedgehog long-range enhancer, the *HOXD* global control region, and the Beta-globin locus control region[49].

Potential target genes were subsequently filtered based on retinal expression. To do so, we retrieved RNA-seq paired-end FASTQ files derived from fetal retina samples (52 to 136 days post-conception) characterized in Hoshino et al.[83]. Transcripts were quantified through pseudo-alignment by *Kallisto* (v.0.46.1)[87] using default parameters for both index build (derived from the Ensembl human release 101) and transcript abundance estimation. Transcript estimates were imported and summarized to create gene-level count matrices using *tximport* (v3.17)[88]. A custom script was then used to filter out target genes exhibiting no expression (TPM < 0.5) at any of the interrogated stages. In addition, to evaluate the reliability of the regulatory domains assigned by *GREAT* to these target genes, we integrated the 2948 retinal TADs described by Marchal et al.[27]. To characterize further the putatively regulated target genes, we made use of the integration of the scATAC-seq gene scores and the scRNA-seq gene expression matrices generated by *ArchR*. In particular, a gene score is a prediction of how highly expressed a gene will be based on the accessibility of nearby regulatory elements[26]. For each target gene, its expression was ranked by percentile of expression across all clusters. A gene expression signature was then assigned by retrieving the cluster identities exhibiting an expression value above the 80th percentile threshold.

In addition, we compared the sets of target genes assigned to UCNEs by *GREAT* and subsequent gene expression filtering, to those mapped using the peak-to-gene linkage method implemented by *ArchR*. To do so, scATAC-seq data peaks that had a Peak2GeneLinkage correlation above 0.4 and their corresponding target genes were kept. UCNES were then assigned to these peaks by *bedtools intersect* as described above, including a ±250-bp-long window The overlap between predicted target genes was computed with respect to those assigned by *GREAT*.

Finally, to identify overrepresented Gene Ontology (GO) terms and infer possible enriched pathways among the potential target genes under UCNE regulation, GO enrichment analyses were performed using *Enrichr*[89].

## Interrogation of WGS data in a rare eye disease cohort

Putatively active UCNEs associated with genes expressed in retina were filtered further based on their implication in disease as retrieved from the comprehensive *gene-disease pairs and attributes* list provided by G2P (Eye and Developmental Disease –DD– Panels; 2022-03-17)[90,91] extended with the NCMD-associated *IRX1* locus[17]. To assess the contribution of genomic variation within these loci to disease, an analysis was performed to detect small variants (SNVs, and indels <50 bp), and large structural variants (SVs) including copy number variants (CNVs) overlapping these disease-gene associated UCNE loci through query of a sub-cohort of participants with rare eye disease phenotypes (*n* = 3206) from the 100,000 Genomes Project (100KGP, Genomics England, of which 1802 individuals were unsolved. Retrieved variants were annotated with VEP (v.107). As a first filter, only variants with minor allele frequency (MAF) < 0.5% and no observed homozygotes in gnomAD v3.1 were considered for further assessment. Variants were evaluated for their potential pathogenic/modifying effect using in

silico prediction tools focusing on transcription factor (TF) binding site disruption—*QBiC-Pred*—and chromatin state effects—*DeepSea*, *CARMEN*, *FATHMM-XF*, *RegulationSpotter*—under default parameters[59–62,92]. Furthermore, we annotated these variants with our integrated analyses in order to evaluate them within their regulatory context and potential target genes.

For each candidate variant, we compared the similarities between the participant phenotype (HPO terms) and the ones known for its target gene through literature search and clinical assessment by the recruiting clinician when possible. Finally, for each candidate variant identified in participants whose cases were not solved through 100KGP, a variant screening of 387 genes listed in either the Retinal disorders panel (v2.195) from Genomics England PanelApp[93] or RetNet (https://sph.uth.edu/retnet/) was performed to discard (likely) pathogenic variants, both SNVs and SVs, that could provide an alternative molecular diagnosis. For each instance for which only the UCNE variant remained as candidate, we placed a clinical collaboration request with Genomics England.

## Evaluation of allele frequency distribution within UCNEs

Allele frequency distributions were created for the set of variants retrieved within disease-gene-associated UCNE loci and compared against the distribution derived from a background composed of 200 random genomic sequences. This background was generated using the *random-genome-fragments* utility of Regulatory Sequence Analysis Tools (*RSAT*) with a fixed length of 350 bp[94]. Importantly, *GREAT* was used to evaluate the distribution of distances of the randomly selected regions to the TSS of genes to ensure no confounding effects in the downstream analyses. Indeed, no significant differences were observed in such distribution as compared to that of the UCNEs (Supplementary Figure 6). Moreover, no GO terms were overrepresented among the target genes assigned to these random genomic regions, thereby further supporting the suitability of this background for the analysis.

To evaluate whether the depletion of common variation within UCNEs is a more general phenomenon, we made use of genome-wide residual variation intolerance scores (gWRVIS), a nucleotide-resolution metric that quantifies genomic constraint. As such, lower gwRVIS values correspond to greater intolerance to variation[28]. We downloaded gwRVIS data (hg38-build, v1.1) for all chromosomes and queried it for the regions of interest using *tabix* (v1.7.2)[95]. More specifically, these regions included: disease-gene-associated UCNE loci, a flanking region encompassing 200 bp up- and downstream of each of these UCNEs, and the 200 random genomic sequences described above. A Kruskal–Wallis test followed by post-hoc Dunn test with a Bonferroni correction for multiple hypothesis testing was used to compare the gwRVIS distributions across all pairs.

In addition, in order to assess the specificity of the overlap of rare variants in the studied patient sub-cohort with UCNEs, we generated allele frequency distributions corresponding to all variants retrieved for a selection of 25 target disease gene loci under putative UCNE regulation and their corresponding UCNEs. For each disease-gene and UCNE pair, we plotted the proportion of ultrarare (absent from gnomAD v3.1), very rare (MAF < 0.1%), rare (0.1% ≤ MAF < 0.5%), low frequency (0.5% ≤ MAF < 5%), and common (MAF ≥ 5%) variants.

## Targeted sequencing and reverse phenotyping

We performed segregation testing of two ultrarare SNVs initially identified in three affected individuals of a 3-generation family displaying autosomal dominant foveal abnormalities, namely an SNV in a candidate *cis*-regulatory UCNE located upstream of *PAX6* and a missense *CFH* variant. Genomic DNA was extracted using Oragene-DNA saliva kits (OG-500, DNA Genotek) according to manufacturer's instructions. Targeted sequencing of the variants was performed on

**Article** https://doi.org/10.1038/s41467-024-45381-1

genomic DNA by PCR amplification followed by Sanger sequencing using the BigDye Terminator v3.1 kit (Life Technologies). Primer sequences can be found in Supplementary Data 12.

Following their genetic assessment, four members of this 3-generation family were clinically re-evaluated. The examination included visual acuity assessment, slit-lamp examination for anterior and posterior segment anomalies, and intraocular pressure measurement. Detailed imaging involving ultra-wide field fundus photography, ultra-wide field autofluorescence imaging and optical coherence tomography (OCT) was performed.

### Generation of in vivo reporter constructs and functional characterization of candidate *cis*-regulatory UCNE upstream of *PAX6* using enhancer assays in zebrafish embryos

Primers were designed to amplify the sequence of the candidate *cis*-regulatory UCNE located upstream of *PAX6* from human genomic DNA (Roche). The PCR product was then cloned into the E1b-GFP-Tol2 enhancer assay vector containing an E1b minimal promoter followed by the Green Fluorescent Protein (GFP) reporter gene[96] by restriction-ligation cloning. The primer sequences can be found in Supplementary Data 12. The recombinant vector containing the cCRE-*PAX6*-UCNE was then amplified in One Shot TOP10 Chemically Competent *E. coli* cells (Invitrogen) and purified using the QIAprep Spin Miniprep Kit (Qiagen). In addition, a construct containing the ultrarare SNV identified in the affected individuals of the family described above was created using the Q5 Site-Directed Mutagenesis Kit (NEB) using variant-specific primers designed with the NEBaseChanger tool. The sequence of the insert was confirmed by Sanger sequencing using the BigDye Terminator v3.1 kit (Life Technologies). The constructs were then micro-injected into the yolk of at least 70 wild-type zebrafish embryos (AB) at single-cell stage using the Tol2 transposase system for germline integration of the transgene according to Bessa et al.[97] with minor modifications; we injected 1.4 nL droplets containing 50 ng/µL transposase mRNA and 40 ng/µL Tol2 plasmid. As readout, GFP fluorescence was observed and photographed with a Leica M165 FC Fluorescent Stereo Microscope (Leica Microsystems) and its localization annotated at 1, 2, and 3 days post fertilization (dpf) to evaluate enhancer activity. Assays on zebrafish were done in agreement with EU Directive 2010/63/EU for animals. All efforts were made to minimize pain and discomfort.

### Reporting summary

Further information on research design is available in the Nature Portfolio Reporting Summary linked to this article.

## Data availability

The data that support the findings of this study are available within the Genomics England (protected) Research Environment but restrictions apply to the availability of these data, as access to the Research Environment is limited to protect the privacy and confidentiality of participants. De-identified data as well as analysis scripts are, however, available from the authors upon reasonable request. Source data are provided with this paper.

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

## Acknowledgements

This work was supported by the Ghent University Special Research Fund (BOF20/GOA/023) (E.D.B.); H2020 MSCA ITN grant (No. 813490 StarT) (E.D.B., F.C., M.B.), EJPRD19-234 Solve-RET (E.D.B.). E.D.B. is a Senior Clinical Investigator (1802220N) of the Research Foundation-Flanders (FWO); V.L.S. and A.D.R. were an Early Starting Researcher of StarT (grant No. 813490). E.D.B. is member of ERN-EYE (Framework Partnership Agreement No. 739534-ERN-EYE). This research was made possible through access to the data and findings generated by the 100,000 Genomes Project. The 100,000 Genomes Project is managed by Genomics England Limited (a wholly owned company of the Department of Health and Social Care). The 100,000 Genomes Project is funded by the National Institute for Health Research and NHS England. The Welcome Trust, Cancer Research UK and the Medical Research Council have also funded research infrastructure. The 100,000 Genomes Project uses data provided by patients and collected by the National Health Service as part of their care and support. Figures from the VISTA enhancer browser were used under permission of the Berkeley Lab (under a CC BY license) that is DOE funded. We acknowledge Chris Inglehearn (Leeds Institute of Medical Research, School of Medicine, University of Leeds, Leeds, UK) for his helpful advice and comments on the manuscript. We also would like to thank the Zebrafish Facility Ghent (ZFG) Core at Ghent University. Hanna De Saffel, Quinten Mahieu, Angelika Jürgens, and Lies Vantomme are thanked for their excellent technical assistance. This publication is part of the Human Cell Atlas - www.humancellatlas.org/publications (HCA-51).

## Author contributions

V.L.S.: Conception and project design, acquisition of data, analysis and interpretation of data, drafting and revising the manuscript. A.D.R.: Conception and project design, acquisition of data, analysis and interpretation of data, drafting and revising the manuscript. R.M.: Acquisition of data, analysis and interpretation of data, revising the manuscript. G.E: Acquisition of data, revising the manuscript. F.C: Acquisition of data, revising the manuscript. M.B.: Project supervision, acquisition of data, revising the manuscript. A.W.: Acquisition of data, analysis and interpretation of data, revising the manuscript. E.D.B.: Conception and project supervision, acquisition of data, analysis and interpretation of data, drafting and revising the manuscript.

## Competing interests

The authors declare no competing interests.

## Additional information

## Genomics England Research Consortium

**Rajarshi Mukherjee**[3] **& Chris F. Inglehearn**[5]

[5]Division of Molecular Medicine, Leeds Institute of Medical Research, University of Leeds, Leeds, UK.

