## [Peer Review File · Nature Communications]

Multi-omics analysis in human retina uncovers ultraconserved cis-regulatory elements at rare eye disease lociReviewer #2 (Remarks to the Author):

The authors have responded to my recommendations in the previous review in an entirely satisfactory manner. I also noticed substantial improvements of the manuscript addressing points raised by the other reviewers.

In my opinion, this is now an excellent paper ready for publication.

Reviewer #3 (Remarks to the Author):

The authors have addressed most of the questions raised by the reviewers and the manuscript is improved. There are two remaining concerns.

1. To identify candidate rare variants at UCNEs, rare variants identified from WGS of 3,220 patients are intersected with UCNEs. However, since pathogenic mutations from a significant portion of these 3,220 IRD patients have been identified, only patients remain unsolved should be used in the search. This should be updated to avoid misleading impression as now the the number of candidate variants within UCNEs is inflated.
2. Given that the UCNE at the Pax6 locus with and without the variant shows similar enhancer activity in zebrafish, it casts doubt on the pathogenicity of the variant and/or the UCNE itself. It needs to be further clarified by results such as either editing the site or deleting the element to see if the base and/or the element is essential for proper Pax6 expression/function. Since the Pax6 case is the only positive example reported in the manuscript, the negative result would make one wonder the effectiveness of the strategy.

RESPONSE TO REVIEWERS' COMMENTS

Reviewer #2 (Remarks to the Author):

The authors have responded to my recommendations in the previous review in an entirely satisfactory manner. I also noticed substantial improvements of the manuscript addressing points raised by the other reviewers.

In my opinion, this is now an excellent paper ready for publication.

Answer: No changes needed.

Reviewer #3 (Remarks to the Author):

The authors have addressed most of the questions raised by the reviewers and the manuscript is improved. There are two remaining concerns.

#1. To identify candidate rare variants at UCNEs, rare variants identified from WGS of 3,220 patients are intersected with UCNEs. However, since pathogenic mutations from a significant portion of these 3,220 IRD patients have been identified, only patients remain unsolved should be used in the search. This should be updated to avoid misleading impression as now the the number of candidate variants within UCNEs is inflated.

Answer:

Thanks again for your valuable feedback. We appreciate your concern about potential misleading impressions arising from the identification of rare candidate variants at UCNEs based on the entire cohort, in which some cases were already characterized/solved.

To address this, we conducted an additional analysis to distinguish between solved and unsolved cases within the cohort studied. After this analysis, we were able to retrieve information for 2,891 participants, for which the solved/unsolved status is the following: Unsolved: 1,802; Solved: 898; Partially solved: 35; Report not available: 31.-; Unknown: for the remaining 440.

In addition, we could retrieve information for those cases for which we have linked the variants listed in Supplementary Table 7 ("178 are located within 84 UCNEs displaying histone modification marks"). Regarding the status of solved/unsolved cases related to the identified variants: Unsolved: 109, Solved: 49, Partially solved: 2, Report not available: 2, Unknown: 16. Notably, the majority of these variants was found in unsolved cases. This information was also included in the main text (Methods, Interrogation of WGS data in a rare eye disease cohort). Specifically: *"The sub-cohort's solved/unsolved status is as follows: unsolved (1,802), solved (898), partially solved (35), report not available (31), and unknown (440). Regarding the solved/unsolved status associated to the identified variants (listed in Supplementary Table 7): unsolved (109), solved (49), partially solved (2), report not available (2) and unknown (16)."*

It was decided to conduct this naive screening for two reasons: (i) For an important fraction of cases (as specified), there is no information regarding the solved/unsolved status. (ii) Apart from a (likely) pathogenic effect, it cannot be excluded that noncoding variants in UCNEs can modify the penetrance or expressivity of (likely) pathogenic (coding) variants, making it relevant to also include 'solved' patients with (likely) pathogenic coding variants in our screen and report the results.

Overall, these clarifications address this concern and provide a more accurate representation of our analysis and the studied cohort.

#2. Given that the UCNE at the Pax6 locus with and without the variant shows similar enhancer activity in

zebrafish, it casts doubt on the pathogenicity of the variant and/or the UCNE itself. It needs to be further clarified by results such as either editing the site or deleting the element to see if the base and/or the element is essential for proper Pax6 expression/function. Since the Pax6 case is the only positive example reported in the manuscript, the negative result would make one wonder the effectiveness of the strategy.

Answer:

Using zebrafish enhancer assays of the *PAX6*-associated UCNE in zebrafish, we demonstrated this UCNE displays reporter activity in the human embryonic forebrain (6/9 embryos) as well as in other structures including eye (3/6 embryos) and neural tube. Although we showed that the *PAX6*-associated UCNE is an active enhancer, we acknowledge it would require further work to test the enhancer activity of the UCNE (f.i. by deleting the element in retinal organoids) and whether variation of the UCNE would lead to differences in *PAX6* expression and to a phenotype.